# SMARCB1 (INI-1)-Deficient Sinonasal Carcinoma: A Systematic Review and Pooled Analysis of Treatment Outcomes

**DOI:** 10.3390/cancers14133285

**Published:** 2022-07-05

**Authors:** Victor Ho-Fun Lee, Raymond King-Yin Tsang, Anthony Wing Ip Lo, Sum-Yin Chan, Joseph Chun-Kit Chung, Chi-Chung Tong, To-Wai Leung, Dora Lai-Wan Kwong

**Affiliations:** 1Department of Clinical Oncology, School of Clinical Medicine, LKS Faculty of Medicine, The University of Hong Kong, Hong Kong, China; csy023@ha.org.hk (S.-Y.C.); tccz01@ha.org.hk (C.-C.T.); leungtw@ha.org.hk (T.-W.L.); dlwkwong@hku.hk (D.L.-W.K.); 2Department of Clinical Oncology, Queen Mary Hospital, Hong Kong, China; 3Department of Surgery, School of Clinical Medicine, LKS Faculty of Medicine, The University of Hong Kong, Hong Kong, China; rkytsang@hku.hk (R.K.-Y.T.); cck000@ha.org.hk (J.C.-K.C.); 4Department of Otorhinolaryngology, Queen Mary Hospital, Hong Kong, China; 5Department of Pathology, Queen Mary Hospital, Hong Kong, China; lwi543@ha.org.hk

**Keywords:** SMARCB1-deficient, INI1-deficient, sinonasal, paranasal, tazemetostat, treatment outcomes, systematic review

## Abstract

**Simple Summary:**

SMARCB1 (INI-1)-deficient sinonasal carcinoma is a rare but locally aggressive malignancy which usually presents very late with invasion into the orbits and intracranium, and compression of adjacent cranial nerves and their branches at diagnosis, and has a poor survival outcome. So far, less than 200 cases have been reported. The optimal treatment strategy for this rare but intractable malignancy is yet to be determined. We performed a systematic review and included patients treated at our institution for treatment outcome and survival analysis. Multivariable analysis revealed that T4b disease at diagnosis was the only prognostic factor of overall survival. In view of the extreme challenge of managing this malignancy, we are initiating a phase II trial on tazemetostat, an EHZ2 inhibitor, as induction therapy with systemic chemotherapy followed by radical surgery or chemoradiation, and as maintenance therapy for previously untreated stage III to IVA SMARCB1 (INI-1)-deficient sinonasal carcinoma.

**Abstract:**

(1) Background: SMARCB1 (INI-1)-deficient sinonasal carcinoma is a rare sinonasal malignancy; since its discovery and description in 2014, less than 200 cases have been identified. It is almost impossible to perform randomized-controlled trials on novel therapy to improve treatment outcomes in view of its rarity. We performed a systematic review of all the published case reports/series and included our patients for survival analysis. (2) Methods: In this systematic review, we searched from PubMed-MEDLINE, EMBASE, Scopus, Cochrane Library, CINAHL, and Google Scholar for individual patient data to identify and retrieve all reported SMARCB1-deficient sinonasal carcinoma. Clarification on treatment details and the most updated survival outcomes from all authors of the published case reports/series were attempted. Survival analysis for overall survival (OS) and identification of OS prognostic factors were performed. This systematic review was registered with PROSPERO (CRD42022306671). (3) Results: A total of 67 publications were identified from the systematic review and literature search. After excluding other ineligible and duplicated publications, 192 patients reported were considered appropriate for further review. After excluding duplicates and patients with incomplete pretreatment details and survival outcomes, 120 patients were identified to have a complete set of data including baseline demographics, treatment details, and survival outcomes. Together with 8 patients treated in our institution, 128 patients were included into survival analysis. After a median follow up of 17.5 months (range 0.3–149.0), 50 (46.3%) patients died. The 1-year, 2-year and 3-year OS rates were 84.3% (95% CI % 77.6–91.0), 62.9% (95% CI 53.1–72.7), and 51.8% (95% CI 40.8–62.8), respectively, and the median OS was 39.0 months (95% CI 28.5–49.5). Males (*p* = 0.029) and T4b disease (*p* = 0.013) were significant OS prognostic factors in univariable analysis, while only T4b disease (*p* = 0.017) remained significant in multivariable analysis. (4) Conclusions: SMARCB1-deficient sinonasal carcinoma is an extremely aggressive sinonasal malignancy with a dismal prognosis. Early diagnosis and a multimodality treatment strategy are essential for a better treatment and survival outcome.

## 1. Introduction

SMARCB1 (INI-1)-deficient sinonasal carcinoma is a rare but locally aggressive malignancy of the nasal cavity and paranasal sinuses, representing only about 1% of all head and neck malignancies [1,2]. SMARCB1 (INI1) is a core subunit of the SWItch/sucrose non-fermentable (SWI/SNF) complex mapped to chromosomal region 22q11.23. Though many SWI/SNF protein subunits are interchangeable, SMARCB1 (also known as INI-1, INI1, or integrase interactor 1, BAF47, SNF5, CSS3, Snr1, MRD15, RDT, RTPS1, Sfh1p, hSNFS, SNF5L1, SWNTS1, and PPP1R144) is an essential component of all SWI/SNF complexes [1]. SWI/SNF complex subunits are responsible for regulating vital processes of cellular differentiation and proliferation through their enrichment at sites of promoters and enhancers of active genes, which possess important roles of tumor suppression [3]. Therefore, the SWI/SNF complex is a constellation of tumor suppressor genes, though the exact function of each component remains undeciphered. So far, the vast majority of SWI/SNF complex-related head and neck malignancies have been associated with either SMARCB1 or SMARCA4 inactivation.

SMARCB1-deficient sinonasal carcinoma was first described and reported in 2014, and was first recognized as a subset of sinonasal undifferentiated carcinoma in the 4th edition of the World Health Organization’s Classification of Head and Neck Tumors in 2017 [1,2,4]. The overwhelming majority of SMARCB1-deficient sinonasal carcinoma presents very late, with locoregionally advanced disease at diagnosis, owing to its very similar clinical presentation to other benign conditions, such as allergic rhinitis, nasal polyps, chronic sinusitis, and some commoner malignancies, such as HPV-associated squamous cell carcinoma, extranodal NK/T cells lymphoma, and mucosal melanoma. Therefore, complete surgical removal remains very challenging, and, most of the time, it is impossible.

So far, only less than 200 patients with SMARCB1-deficient sinonasal carcinoma have been reported. Its treatment strategy is mainly extrapolated from management of other types of aggressive sinonasal carcinoma. Upfront radical resection followed by adjuvant radiation therapy or adjuvant chemoradiation is usually the standard of care for potentially resectable disease, taking reference from the management of other sinonasal malignancies of differential histological types [5,6,7,8]. On the other hand, induction chemotherapy followed by surgery and adjuvant radiation was also explored [9], and is being evaluated in a phase II randomized-controlled trial (RCT) (NCT03493425) as well [10]. A universally accepted and recommended treatment guideline for this dreadful malignancy is hitherto still lacking. In view of this, we performed a systemic review and evaluated the treatment details and survival outcomes of all reported cases. With the inclusion of treatment details and survival outcomes of patients managed at our institution, survival analyses and identification of poor prognostics of overall survival (OS) were conducted and reported herewith.

## 2. Materials and Methods

### 2.1. Search Strategy and Selection Criteria

We performed a systematic review and literature search, in accordance with the Preferred Reporting Items for Systematic Review and Meta-analysis (PRISMA) guidelines for all publications on all histologically or cytologically confirmed SMARCB1-deficient sinonasal malignancies (Figure 1 and Appendix A). PubMed-MEDLINE, Embase (Ovid), Scopus, Cochrane Library, CINAHL, and Google Scholar were searched using the search terms “sinonasal OR paranasal sinus OR paranasal sinuses OR nasal”, and “SMARCB1 OR SMARCB-1 OR INI1 OR INI-1 OR INI?”. The full strategy can be found in the Appendix A. Our search focused on all full-length articles, case reports, and case series published in the English literature. The search date started from the inception of these databases to 31 March 2022. Abstract and full text review were carried out by two authors (V.H.-F.L. and R.K.-Y.T.) independently. Discrepancy was resolved by consensus. For any duplicated, unknown, missing, or non-updated information on baseline patient disposition, treatment details, survival status, and survival outcomes in the published articles, we sent emails to the corresponding authors for further clarification in order to avoid repeated patient count, unreliable patient demographic and treatment characteristics, and inaccurate survival status and duration. We also identified patients with SMARCB1-deficient sinonasal carcinoma treated at our institution (HKU) since January 2014, with the database locked on 31st March 2022, and included them in subsequent statistical analysis. We registered our systematic review in PROSPERO (CRD42022306671). Approval from the Institutional Review Board of The University of Hong Kong/Hospital Authority Hong Kong West Cluster (reference number UW 22-212) for this study was also obtained before study commencement. The protocol is available in the Appendix A.

### 2.2. Survival Analysis

We evaluated the OS of all patients from the systematic review, as well as of patients treated at our institution, which was defined from the date of histological or cytological diagnosis of SMARCB1-deficient sinonasal carcinoma to the date of death of any cause by Kaplan–Meier methods. Comparison of OS between subgroups was made using log rank tests. Univariable and multivariable analyses of prognostic factors of OS was performed by Cox proportional hazard models. All statistical analyses were performed by Statistical Package for Social Sciences (SPSS) version 27.0. A two-sided *p* value of < 0.05 indicated statistical significance.

## 3. Results

### 3.1. Search Strategy Results

A total of 67 publications was identified from the literature search and systematic review (Figure 1 and Appendix A). After excluding other ineligible and irrelevant publications (Appendix A), 34 publications consisting of 205 patients were considered appropriate for further detailed review [1,2,11,12,13,14,15,16,17,18,19,20,21,22,23,24,25,26,27,28,29,30,31,32,33,34,35,36,37,38,39,40,41,42]. A total of thirteen patients reported in eight publications were confirmed to be duplicates [1,2,12,15,18,21,23,30], so they were just counted once, and their most recent survival outcomes were included in subsequent analysis.

Of the remaining 192 patients, 72 patients whose individual baseline patient characteristics, treatment details, and survival outcomes could not be identified and verified [2,16,19,20,23,28,29,30,34,35,38], and they were thus not included into our survival analysis. As a result, only 120 patients with data availability of baseline patient characteristics, disease stage, and survival outcomes in 25 publications were included for survival and prognostic factor analyses [1,2,11,13,14,15,16,17,19,21,22,23,24,25,26,27,31,32,33,36,37,39,40,41,42]. Meanwhile, our institution had eight patients who had a complete set of baseline characteristics, treatment details, and survival outcomes, and who were treated between 2014 and 2021. Eventually, altogether, 128 patients were available for subsequent survival analysis; their baseline dispositions are shown in Table 1 and Appendix A.

### 3.2. Histology and Cytomorphology of Tumor Samples

The tumor morphology appeared diverse and variable (Table 2). Detailed histological and cytomorphological description of the tumors could be confirmed in 192 patients. Basaloid with or without other cytomorphological features constituted the most predominant morphological appearance (35.4%). Plasmacytoid/rhabdoid with or without other features and undifferentiated carcinoma were the second and third most common (19.3% and 7.8%, respectively).

### 3.3. Deletional Status of SMARCB1 by Further Molecular and Genetic/Genomic Diagnostics

The tumors of all 128 patients were confirmed to have loss of expression of SMARCB1 (INI-1) with immunohistochemistry. Additional molecular and genetic/genomic diagnostic tests were performed to confirm the status of SMARCB1 deletion in twenty-four patients: fluorescence in-situ hybridization (FISH) in nine patients [2,41], next generation sequencing (NGS) in two patients (including one from our institution) [25], NGS and FISH in one patient [11], NGS and chromogenic in-situ hybridization in four patients [19], and NGS and multiplex ligation-dependent probe amplification in eight patients [36]. Out of these twenty-four patients, monoallelic deletion was identified in three patients [2,36,41], and biallelic deletion was found in nineteen patients [2,11,19,25,36]. Normal results of FISH were noted in the remaining two patients [2].

### 3.4. Treatment Outcomes of Patients in Our Institution

Our institution treated eight patients with SMARCB1-deficient sinonasal carcinoma between January 2014 and December 2021, with the database locked on 31st March 2022 (Appendix A). They were five males and three females, with a median age of 48 years old (range 30–63). One patient had T3 disease, while the rest had T4 disease at diagnosis after baseline workup with magnetic resonance imaging (MRI) of the head and neck and positron-emission tomography with integrated computed tomography (PET/CT). None of them had regional nodal spread or distant metastasis. One patient received upfront radical resection followed by postoperative proton therapy. Another patient underwent radical resection followed by postoperative chemoradiation with 3-weekly cisplatin and proton therapy. Four patients received three cycles of induction chemotherapy with docetaxel, cisplatin, and 5-FU (TPF)—except one patient who received capecitabine to replace 5-FU to avoid prolonged hospitalization in the midst of the COVID-19 pandemic—followed by radical surgery and postoperative chemoradiation with photons and two cycles of 3-weekly cisplatin up to 60–66 Gy in 30–33 fractions over 6–6.5 weeks. One patient received induction chemotherapy with TPF for three cycles, followed by radical chemoradiation with photons in view of the inoperable tumor with very close proximity to the optic nerves and chiasma, albeit with mild tumor shrinkage following induction TPF. The remaining patient, with a complete genomic loss of SMARCB1 confirmed by next-generation sequencing (FoundationOne^®^ CDx), received three cycles of induction TPF followed by concurrent chemoradiation with photons and two cycles of 3-weekly cisplatin (Appendix A). A radiologically complete response was achieved, and the patient was still free from relapse 3.5 years after diagnosis (Appendix A). Two patients developed relapse, including one with local recurrence who was still alive, and another with distal intraspinal extramedullary metastases who died 4 months after relapse. Another patient died of sudden cerebral infarction 9 months after diagnosis and 3 months after radical treatment.

### 3.5. Survival Analysis

After a median follow up of 17.5 months (range 0.3–149.0) (17.0 months (range 0.3–149.0) for 120 previously reported patients, and 30.8 months (range 8.9–50.1) for 8 patients in our institution), 57 patients (44.5%) died, including 2 from our institution. Of them, tumor progression of SMARCB1-deficient sinonasal carcinoma was the most common cause of death (44 patients, 77.2%), followed by unknown causes (7 patients, 12.3%), and cerebrovascular events (6 patients, 10.5%).

The 1-year, 2-year, and 3-year OS rates of all 128 patients in the whole study population, including those treated at our institution, were 84.3% (95% CI 77.6–91.0), 62.9% (95% CI 53.1–72.7), and 51.8% (95% CI 40.8–62.8), respectively, and their median OS was 39.0 months (95% CI 28.5–49.5) (Figure 2a). The corresponding 1-year, 2-year, and 3-year OS of 8 patients treated in our institution were all 75.0% (95% CI 45.0–100), while the rates of the remaining 120 reported patients were 85.1% (95% CI 78.2–92.0), 61.6% (95% CI 51.2–72.0), and 49.6% (95% CI 38.0–61.2), respectively. The median survival of our 8 patients was not reached, and not statistically significant from that of the 118 reported patients (33.0 months, 95% CI 20.9–45.1; *p* = 0.248) (Appendix A). Male patients had a significantly shorter OS (30.0 months, 95% CI 17.1–42.9) compared to the females (median 81.4 months, 95% CI 31.1–131.7; *p* = 0.029) (Figure 2b). In addition, patients with T4b disease had a shorter OS (26.0 months, 95% CI 12.8–39.2) than those with earlier T-category disease (71.0 months, 95% CI 36.6–105.4; *p* = 0.013) (Figure 2c and Appendix A).

### 3.6. Prognostic Factors

Univariable analysis revealed that male sex (*p* = 0.029) and T4b disease (*p* = 0.013) were poor prognostic factors of OS, but only T4b disease remained prognostic in multivariable analysis (*p* = 0.017) (Table 3).

## 4. Discussion

Here, we have presented the largest systematic review of patient characteristics, treatment details, and survival outcomes of patients with SMARCB1-deficient sinonasal carcinoma, among the two previously published which did not include all reported cases [29,42]. SMARCB-1 deficient sinonasal carcinoma, first reported in 2014, remains one of the rarest and most poorly understood sinonasal malignancies. The gene products of SMARCB1 are abundantly expressed in the nuclei of all normal human cell types [3], similar to most of the other components of the SWI/SNF complex. Biallelic inactivation of the gene leads to complete loss of SMARCB1 expression, which can be routinely detectable by immunohistochemical staining as the standard diagnostic criteria of SMARCB1-deficient malignancy [43]. Most of the cases are sporadic, while germline mutations have also been reported. Recently, comprehensive genomic profiling using next-generation sequencing has been increasingly employed in diagnosing this malignancy, including one patient in our institution, showing loss of SMARCB1 and absence of other genomic aberrations. The tumors of these patients usually had a stable microsatellite status and a low tumor mutation burden. However, the histological features of SMARCB1-deficient sinonasal malignancy vary significantly, which are most commonly basaloid and plasmacytoid/rhabdoid, followed by squamous, squamous papillary, glandular (appearing as non-intestinal adenocarcinoma), clear-cell type, and yolk-sac tumor differentiation, as reported previously and in our current study [16,43]. Quite often, SMARCB1-deficient sinonasal carcinoma presents as small blue round cell tumors which mimic other sinonasal malignancies including undifferentiated carcinoma, NUT sinonasal carcinoma, lymphoma, small cell carcinoma, olfactory neuroblastoma, melanoma, and rhabdomyosarcoma. Immunohistochemical staining, such as synaptophysin, chromogranin, SALL4, etc., and further molecular investigations, including in-situ hybridization for Epstein–Barr virus-encoded early RNA or even relevant translocation studies, are essential for working up these histologically diverse tumors [3]. Similar to other histological and molecular types of sinonasal malignancies, its clinical presentation is often local or locoregionally advanced with T4a/T4b disease at diagnosis, rendering upfront complete resection very technically challenging. Its overall prognosis, however, is more dismal when compared to SMARCB1-retained, IDH2 R172-mutant, and IDH2 wild-type counterparts [28,33,36,44].

Although there seems to be a general consensus on employing multimodality therapy, the exact sequence of multimodality is highly variable and diversified, which is primarily based on the expertise of the treating institutions [45,46,47]. Upfront surgical resection for this disease has been advocated by many centers owing to its rarity and advanced and aggressive clinical presentation [5,6,7,8,47]. The survival outcomes, however, remained very disappointing, except for a few patients who might have long-term survival after complete clearance with multimodality treatment. In view of the paucity of evidenced-based treatment strategies for this extremely rare and underreported malignancy, we performed a systemic review and evaluated the treatment details and survival outcomes of all reported cases, and hoped that an acceptable current and future treatment strategy could be proposed and devised. The recent retrospective study on the response to up to three cycles (ranging from one to five) of induction chemotherapy with docetaxel, cisplatin, and etoposide as a guide to optimize the subsequent radical treatment modality has roughly shaped the treatment paradigm for sinonasal undifferentiated carcinoma [9]. Those who had a favorable response to induction chemotherapy and subsequently received radical chemoradiation enjoyed longer disease-specific survival (DSS) than those who received radical surgery. On the other hand, those who achieved a less favorable response to induction chemotherapy had longer DSS after radical surgery when feasible. These findings underscored the importance of testing the chemosensitivity (and probably radiosensitivity) of this disease, which may also pave the way for the choice of subsequent treatment. That said, the balance between efficacy and chemotherapy-related toxicities must be well evaluated and assessed on an individual basis in multidisciplinary meetings with surgeons and oncologists. Alternative treatment options have to be contemplated and well planned when induction chemotherapy is not effective or resulting in unacceptable toxicities.

Based on the experience accumulated in our institution, we would offer induction chemotherapy with TPF for three cycles for all physically fit patients with non-metastatic T4 SMARCB1-deficient sinonasal carcinoma, followed by immediate imaging assessment with an MRI and CT scan of the head and neck region (Appendix A). If tumor shrinkage of greater than 50% is observed after induction TPF, which implies a favorable chemosensitivity, patients should receive radical concurrent chemoradiation up to 66–70 Gy with two to three cycles of platinum compounds every 3 weeks. For those with less than 50% tumor shrinkage, patients should receive radical surgery aiming at macroscopic and microscopic clearance, followed by post-operative chemoradiation with 60–66 Gy concurrent with two to three cycles of platinum compounds. A higher radiation dose can be considered if patients are treated with charged-particle therapy, such as protons or carbon ions, which has a higher relative biological effectiveness. Clinical follow-up and reassessment imaging with MRI or CT or the head and neck regions, and CT of the thorax and abdomen (or PET/CT) every 3 to 4 months for the first year, then every 4 to 6 months for the second year, every 6 to 9 months for the third year, every 9 to 12 months for the fourth year, and yearly from the fifth year onwards are recommended. For metastatic disease at diagnosis, treatment would be palliative platinum-based chemotherapy, with or without palliative surgery or radiation therapy for better local disease control.

Multimodality and multidisciplinary management of this intractable malignancy is the current mainstay of treatment. Upfront surgery followed by postoperative adjuvant radiation therapy or concurrent chemoradiation has been recommended as the standard of care for potentially resectable disease. The alternative would be induction chemotherapy followed by surgery and/or (chemo)radiation, and its efficacy and safety is being evaluated in an ongoing phase II RCT (NCT03493425) comparing upfront surgery followed by postoperative (chemo)radiation therapy to induction docetaxel and cisplatin for up to three cycles followed by surgery and postoperative (chemo)radiation therapy, which recruits paranasal sinus squamous cell carcinoma but is not limited to SMARCB1-deficient tumors [10]. Nevertheless, recurrence rates are generally high, and further novel therapies are warranted. Immune checkpoint inhibitors have been gaining increasing acceptance in treating head and neck squamous cell carcinoma (HNSCC), as exemplified in the recently published phase III RCT (KEYNOTE-048) study (NCT02358031) on first-line pembrolizumab and chemotherapy vs. the EXTREME regimen of cetuximab and chemotherapy for recurrent/metastatic HNSCC [48]. However, very limited data is available on its efficacy against SMARCB1-deficient sinonasal carcinoma. The very few ongoing clinical trials which are recruiting very rare tumors including SMARCB1-deficient cancers (NCT02834013) and recurrent/metastatic HNSCC (NCT03370276) are now under way [49,50]. Given its microsatellite stability and low tumor mutation burden reported in previous publications and our patients, the response to immune checkpoint inhibitors is expected to be suboptimal.

Identification of molecular biomarkers and new targeted therapies are essential for clinicians to devise novel personalized treatments for this disease. Previously, preclinical and clinical studies have revealed that SWI/SNF tumor suppressor proteins act as antagonists of the polycomb gene enhancer of zesta homolog 2 (EZH2), a catalytic subunit of the PRC2 polycomb complex which mediates histone methylation, leading to silencing of tumor suppressor genes, metastasis, and conferring drug resistance [51,52,53,54]. Absence of the gene product of SMARCB1 alters the SWI/SNF complex function and leads to increased EZH2 activity, which, in turn, upregulates oncogenic pathways, for instance myc, sonic hedgedog, WNT/b-catenin, and suppresses tumor suppressor gene transcription. Tazemetostat (EPZ-6438), a potent EHZ2 inhibitor, has emerged itself as a potent targeted agent for SMARCB1-deficient malignancies [53]. A number of phase I and II open-label studies have demonstrated that tazemetostat has brought promising treatment outcomes and manageable toxicity profiles in various types of cancers, including diffuse large B-cell lymphoma, follicular lymphoma, epithelioid sarcoma, synovial sarcoma, rhabdoid tumor of the ovary, and peripheral nerve sheath tumors (NCT01897571, NCT02601950, NCT02601937, NCT03213665, NCT04917042) [55,56,57,58,59,60]. Other genes/factors and pathways, including GLI1, CCNA1, CDKN2A, EGFR, and FGFR, are also found to have a close association with loss of SMARCB1. Inhibitors of some of these gene/factors and pathways are now being evaluated as well. In view of the potential anti-tumor activity of tazemetostat against SMARCB1-deficient tumors, our institution is initiating a single-arm open-label phase II trial on induction chemotherapy TPF and tazemetostat, followed by either radical surgery and postoperative (chemo)radiation or radical concurrent chemoradiation and maintenance tazemetostat for locally advanced SMARCB1-deficient sinonasal carcinoma (NCT05151588) (Appendix A) [61].

Use of charged-particle therapy with enhanced RBE compared to photon therapy may also improve locoregional control and spare adjacent critical structures, especially the optic nerves/chiasma and brainstem, from radiation-induced toxicities. Retrospective studies on using proton therapy and carbon ion therapy for sinonasal carcinoma including a patient with SMARCB1-deficient sinonasal carcinoma could achieve a local and regional control rate of about 85%, with very limited grade 3 or 4 toxicities [62,63,64,65,66,67].

Several limitations of this study have been identified. The retrospective nature of this review is obviously one of the inevitable limitations, given the extreme rarity of this malignancy reported so far. The inability to retrieve the most updated treatment outcomes, relapse patterns, and survival status of the reported patients from all institutions around the world might affect the interpretation of treatment and survival outcomes, as well as the results of prognostic factors in univariable and multivariable analyses. In addition, the survival status and the cause of death of a few patients in some publications could not be ascertained during our review, since most of the case reports published by pathologists focused primarily on the diagnostic challenges, as well as the molecular and genomic features of SMARCB1-deficient tumors, rather than the clinical outcomes. Our review would be optimal if other survival outcomes including relapse-free survival and cancer-specific survival could also be reported. Having said that, we have already exhausted our maximal efforts amid the raging widespread COVID-19 pandemic to contact the corresponding authors who have kindly shared with us their patients’ outcomes and survival data, which has allowed us to generate the largest cohort of patients with this disease. Lastly, the lack of a standardized management protocol or treatment recommendation and guidelines, leading to variable treatment and survival outcomes, may also render interpretation of survival outcomes difficult and challenging. Our findings must be further validated in future prospective multicenter studies. An urgent and unmet need to identify predictive biomarkers of treatment response of this rare disease is also evident.

## 5. Conclusions

In the present largest comprehensive systematic review and pooled analysis of patients treated at our institution, we identified that T4b disease was a significant poor prognostic factor in patients with SMARCB1-deficient sinonasal carcinoma, which conferred worse survival compared to earlier T-category diseases. Future concerted efforts on more accurate and earlier diagnosis distinguishing from other entities of sinonasal malignancies, multimodality management, and evaluation of novel therapies are compellingly needed to improve its treatment outcomes and survival.

## Figures and Tables

**Figure 1 cancers-14-03285-f001:**
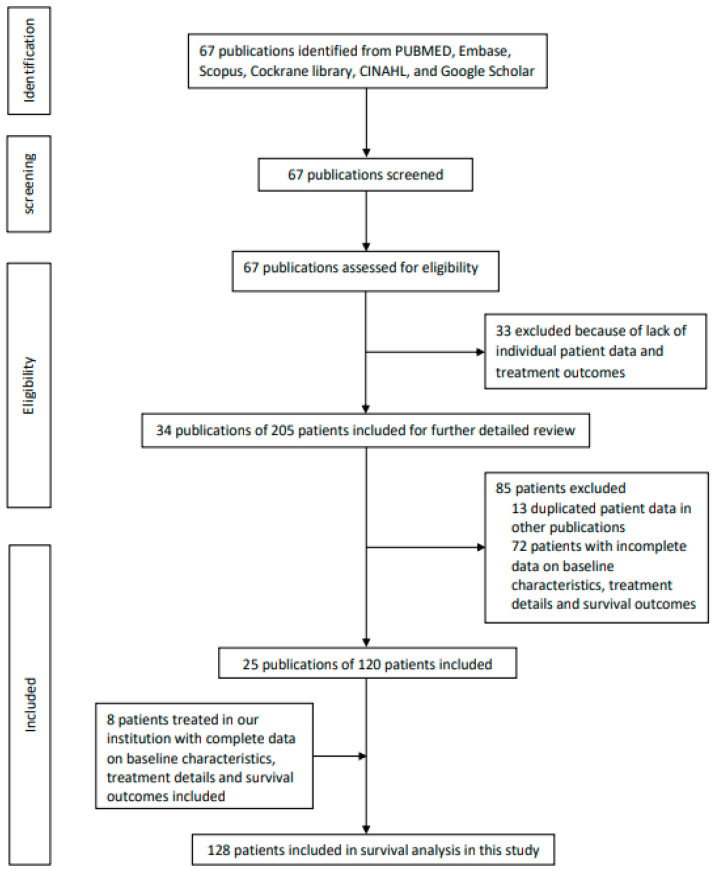
PRISMA flow chart showing the identification and selection of included publications and inclusion of patients treated in our institution for analysis.

**Figure 2 cancers-14-03285-f002:**
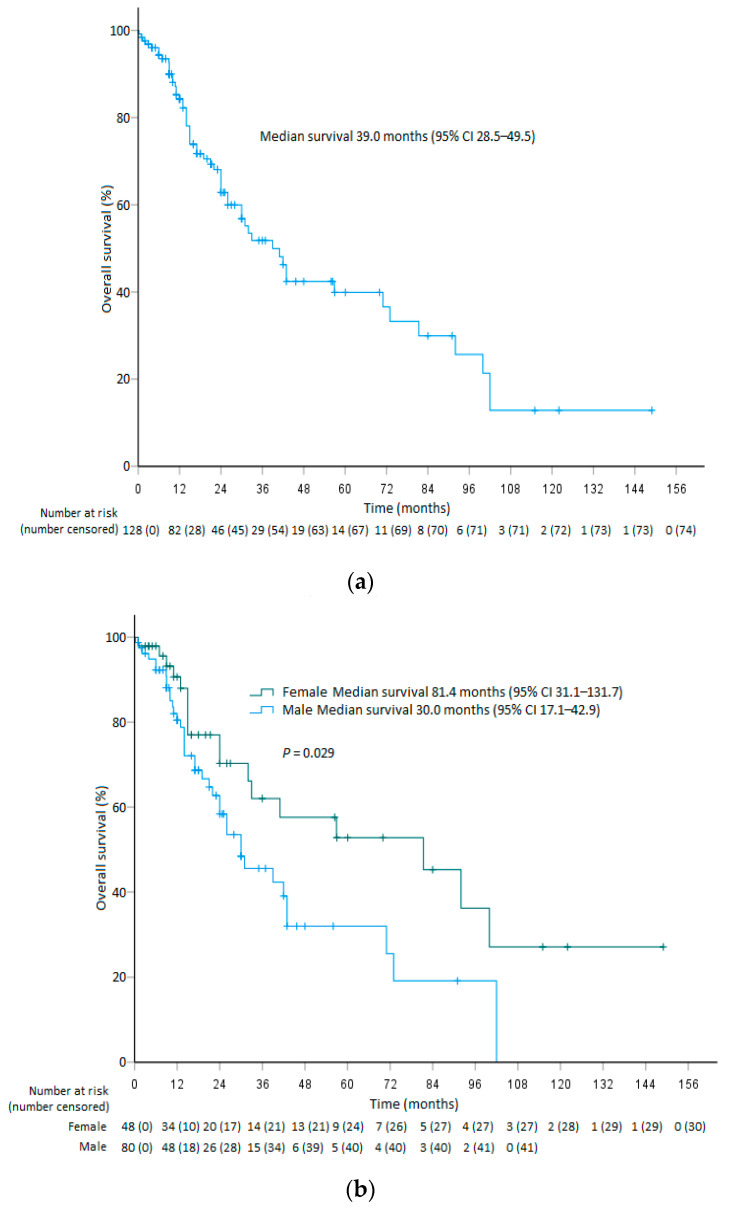
Kaplan–Meier estimates of overall survival showing (**a**) the whole study population (*n* = 128), and the results stratified by (**b**) sex, and (**c**) T1-4a vs. T4b disease.

**Table 1 cancers-14-03285-t001:** Baseline patient disposition.

Characteristics	No. of Patients (%) Total (*n* = 128)
Median age, years (range)	53 (20–89)
Sex	
Male	80 (62.5)
Female	48 (37.5)
T category (Appendix A for definition)	
T1	5 (3.9)
T2	6 (4.7)
T3	14 (10.9)
T4a	53 (41.4)
T4b	50 (39.1)
N category (Appendix A for definition)	
N0	104 (81.3)
N1	4 (3.1)
N2	4 (3.1)
Undetermined	16 (12.5)
M category	
M0 (no distant metastasis)	105 (82.0)
M1 (distant metastasis)	7 (5.5)
Undetermined	16 (12.5)
Radical resection performed	79 (63.3)
Induction treatment received	
Yes	26 (20.3)
No	87 (68.0)
Undetermined	15 (11.7)
Adjuvant treatment received	
Yes	72 (56.3)
No	41 (32.0)
Undetermined	15 (11.7)
Multimodality treatment received	
Yes	96 (75.0)
No	17 (13.3)
Undetermined	15 (11.7)
Only best supportive care received	1 (0.8)

**Table 2 cancers-14-03285-t002:** Distribution of predominant microscopic morphological features.

Predominant Microscopic Morphological Features	No. of Patients (%) Total (*n* = 192)
Basaloid ± other features	
Basaloid only	60 (31.3)
Basaloid, spindled and adenoid	1 (0.5)
Basaloid and focal clear cells	1 (0.5)
Basaloid and squamoid	2 (1.0)
Basaloid and rhabdoid	4 (2.1)
Plasmacytoid/rhabdoid ± other features	
Plasmacytoid/rhabdoid	32 (16.7)
Plasmacytoid/rhabdoid and squamoid	1 (0.5)
Plasmacytoid/rhabdoid and adenoid	1 (0.5)
Plasmacytoid/rhabdoid and glandular differentiation	1 (0.5)
Plasmacytoid/rhabdoid and focal clear cells	1 (0.5)
Plasmacytoid/rhabdoid and oncocytoid	1 (0.5)
Undifferentiated carcinoma	15 (7.8)
Squamous cell carcinoma	7 (3.6)
Adenocarcinoma	6 (3.1)
Pseudoglandular differentiation	4 (2.1)
Squamoid differentiation	3 (1.6)
Small cell carcinoma	1 (0.5)
Poorly differentiated carcinoma	1 (0.5)
Sarcomatoid differentiation	1 (0.5)
High-grade mixed germ cell tumor	1 (0.5)
Yolk sac differentiation	1 (0.5)
Pseudoalveolar and pseudoglandular differentiation	1 (0.5)
Undetermined	46 (24.0)

**Table 3 cancers-14-03285-t003:** Univariable and multivariable analyses of variables prognostic of overall survival.

Covariate	Univariable Analysis			Multivariable Analysis *		
	HR	95% CI	*p*	HR	95% CI	*p*
Age (every 1-year increment)	1.012	0.995–1.030	0.180	-	-	-
Sex (female as reference)	1.856	1.051–3.276	0.029	1.808	0.981–3.302	0.061
T category (T1 as reference)						
T2	1.001	0.008–12.879	0.968	-	-	-
T3	2.012	0.604–6.711	0.255	-	-	-
T4a	1.706	0.696–4.184	0.243	-	-	-
T4b	1.869	1.030–3.390	0.013	1.952	1.134–3.358	0.017
N category (N0 as reference)						
N1	1.553	0.371–6.503	0.547	-	-	-
N2	1.439	0.240–8.614	0.690	-	-	-
M1 disease (M0 as reference)	2.018	0.277–14.717	0.489	-	-	-
Induction treatment	0.925	0.464–1.842	0.824	-	-	-
Adjuvant treatment	0.946	0.498–1.797	0.866	-	-	-
Multimodality treatment	1.260	0.530–2.994	0.601	-	-	-

* Only covariates found significant (*p* < 0.1) in the univariable analysis were considered in the multivariable analysis.

## Data Availability

All articles in this manuscript are available from MEDLINE, Embase, Cochrane Library, CINAHL, and Google Scholar.

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
