# Peer review of "SMARCB1 (INI-1)-Deficient Sinonasal Carcinoma: A Systematic Review and Pooled Analysis of Treatment Outcomes"

_cancers, 2022, doi:10.3390/cancers14133285_

Round 1
Reviewer 1 Report
I appreciate the opportunity to review this well written systematic review of the literature surrounding SMARCB1-Deficient sinonasal carcinoma. Broadly, this is a very well organized and impactful review. It is the largest review of these rare cancers in the literature and provides the most comprehensive analysis of any literature I have seen to date. I think this will be a highly valuable contribution to the literature around this rare malignancy. I have just a few minor comments for the authors to consider. Some major findings to highlight from this paper are key insights into the mysterious biology of this disease. Specifically the review on the activity of SMARCB1 and the SWI/SNF complex is valuable, and the description of the molecular justification for starting a trial around Tazemetostat is commendable and I am very excited to hear the details of this moving forward. This study also highlights the rarity of nodal metastatic disease and highly variable morphologic features that can be seen, so long as INI-1 is lost.
1) I think the manuscript could benefit from a small english language/syntax edit. There are no major concerns, but some of the context and phrasing are awkward throughout the paper, though it does not detract from the content.
2) Figure 1 could be labelled with more clarity, including letters on each component and references to the arrows to better highlight what the reader is seeing with the IHC staining and the INI-1 loss in the tumor cells compared to the stroma.
3) The one comment that I think could be explored more by the authors is their recommendation for induction chemo as a selection protocol for definitive treatment. While this is becoming standard of care for SNUC (where response rates are 80%) it is far less clear of the utility in this cancer. It would be great to see what the response rate to induction chemo is from this literature review (if feasible). TPF is a fairly toxic regimen and if, for example. 50% of patients don't respond, then these patients may not be medically fit enough to undergo radical resection after 3 cycles of induction and if the tumor has grown, it may no longer be resectable. As the authors highlight, there is no evidence that induction is improving survival in these patients right now, so I believe the recommendation for induction is reasonable but perhaps should be provided with more caution or balance on the pros and cons.
I really appreciate the opportunity to review this paper and commend the authors on a great review and well written manuscript.
Author Response
Thank you very much indeed for your review and appraisal of our manuscript. We would like to provide our point-to-point responses to your comments as follows:
I appreciate the opportunity to review this well written systematic review of the literature surrounding SMARCB1-Deficient sinonasal carcinoma. Broadly, this is a very well organized and impactful review. It is the largest review of these rare cancers in the literature and provides the most comprehensive analysis of any literature I have seen to date. I think this will be a highly valuable contribution to the literature around this rare malignancy. I have just a few minor comments for the authors to consider. Some major findings to highlight from this paper are key insights into the mysterious biology of this disease. Specifically the review on the activity of SMARCB1 and the SWI/SNF complex is valuable, and the description of the molecular justification for starting a trial around Tazemetostat is commendable and I am very excited to hear the details of this moving forward. This study also highlights the rarity of nodal metastatic disease and highly variable morphologic features that can be seen, so long as INI-1 is lost.
Our reply: Thank you very much indeed for your great comments and appreciation of our manuscript. We sincerely hope that our revised manuscript which is modified based on the reviewers’ comments is also acceptable.
1) I think the manuscript could benefit from a small english language/syntax edit. There are no major concerns, but some of the context and phrasing are awkward throughout the paper, though it does not detract from the content.
Our reply: Thank you very much indeed for your comments on the manuscript. We have revised the manuscript and invited a nature English speaker to proofread it before re-submission. We sincerely hope that the revised manuscript is more acceptable and smooth.
2) Figure 1 could be labelled with more clarity, including letters on each component and references to the arrows to better highlight what the reader is seeing with the IHC staining and the INI-1 loss in the tumor cells compared to the stroma.
Our reply: Thank you very much for your great comments. We have added letters to each panel of the figure and the figure legend in Supplementary Figure S1 which also clearly indicates the area of immunohistochemical staining and loss of SMARCB1 (IN1-1). We sincerely hope that it is more readable to the reviewer and the readers.
3) The one comment that I think could be explored more by the authors is their recommendation for induction chemo as a selection protocol for definitive treatment. While this is becoming standard of care for SNUC (where response rates are 80%) it is far less clear of the utility in this cancer. It would be great to see what the response rate to induction chemo is from this literature review (if feasible). TPF is a fairly toxic regimen and if, for example. 50% of patients don't respond, then these patients may not be medically fit enough to undergo radical resection after 3 cycles of induction and if the tumor has grown, it may no longer be resectable. As the authors highlight, there is no evidence that induction is improving survival in these patients right now, so I believe the recommendation for induction is reasonable but perhaps should be provided with more caution or balance on the pros and cons.
Our reply: Thank you so much indeed for your nice and valuable comment. We fully agree with the reviewer that induction chemotherapy or induction treatment is commonly used in sinonasal undifferentiated carcinoma and probably in INI1-deficient sinonasal carcinoma. We have searched all the publications and case reports in this systematic review again. Unfortunately, the response rate after induction chemotherapy was rarely mentioned or described, let alone the objective response according to RECIST 1.1. As we are aware, all the publications/case reports included in our systematic review are retrospectively described, and a substantial number of reports primarily described the histology and the protein/genetic/genomic loss of SMARCB1 and just briefly mentioned the clinical course of the patients’ INI1-deficient sinonasal carcinoma. According to the important findings on the retrospective study of 95 patients with previously untreated sinonasal carcinoma published by Amit et al (reference 9 cited in our manuscript), induction chemotherapy can help surgeons and oncologists to formulate the most appropriate and personalized treatment for locally advanced sinonasal carcinoma. In this publication, patients who had a favorable response to induction chemotherapy and subsequently received radical chemoradiation enjoyed a longer disease-specific survival (DSS) than those who received radical surgery. On the other hand, those who achieved a less favorable response to induction chemotherapy had a longer DSS after radical surgery when feasible. Based on this important finding, we opined that induction chemotherapy can aid us to test the chemosensitivity and probably radiosensitivity of this intractable malignancy which may also pave the way for the choice of subsequent therapy. With better supportive care and the more lenient use of granulocyte colony stimulating factor and prophylactic antibiotics after induction chemotherapy with TPF, patients’ toxicity profiles are much more favorable and easily managed. This can be shown in our institution’s cohort that none of our patients developed grade ≥3 toxicity after induction TPF.
In this regard, we have added a separate note in the Discussion section about the careful evaluation and balance between the efficacy and safety of induction chemotherapy in a multidisciplinary meeting with surgeons and oncologists, and hope that it is more acceptable to the reviewer.
We sincerely hope that our elaboration and revision of our manuscript is acceptable to the reviewer. Thank you very much once again for your expert review.
Reviewer 2 Report
Manuscript entitled „SMARCB1 (INI-1)-Deficient Sinonasal Carcinoma: A Systematic Review and Pooled Analysis of Treatment Outcomes” is a very thorough review of literature with added data about the patients from the authors’ hospital. As it is systematic review of almost all so far recognized cases of rare disease it is of great value for the broader audience that might encounter such patient.
Both the main text and supplementary material provide detailed data about the way the study was conducted and gained results.
I have only few minor suggestions:
- In simple summary it would be good to define locally aggressive and T4b disease
- In the summary and introduction, it is stated that so far 100 cases were found, but authors perform study on 128 cases – it would be helpful to clarify this
- On the same note, in Table 2 and in whole chapter about histological findings authors describe 192 patients, but study includes only 128 – I suggest to include in this chapter/table only patients that are a part of the study this manuscript uses (128)
- In table 1 it would be helpful to define meaning of different categories listed in the table
- Although authors included relatively high number of their own cases (8) they were collected in a period that end in March 2022 - cases diagnosed in last few (e. g. 6) months can hardly have useful follow up
- OS was defined by time between diagnosis and death by any causes – I strongly suggest to define it as period between diagnosis and death related to SMARCB1 (INI-1)-Deficient Sinonasal Carcinoma
- It would also be beneficial to have information how was, in different studies, SMARCB1 deficiency established and if deletion was biallelic
- I don't see reason for the inclusion of phase II trial initiation in this manuscript – it doesn't add anything to the topic of the study since it is not jet performed
Author Response
Thank you very much indeed for your review and appraisal of our manuscript. We would like to provide our point-to-point responses to your comments as follows:
Manuscript entitled „SMARCB1 (INI-1)-Deficient Sinonasal Carcinoma: A Systematic Review and Pooled Analysis of Treatment Outcomes” is a very thorough review of literature with added data about the patients from the authors’ hospital. As it is systematic review of almost all so far recognized cases of rare disease it is of great value for the broader audience that might encounter such patient.
Both the main text and supplementary material provide detailed data about the way the study was conducted and gained results.
I have only few minor suggestions:
- In simple summary it would be good to define locally aggressive and T4b disease
Our reply: Thank you very much for your nice comment. We have modified our simple summary with further elaboration that these locally aggressive tumor usually presents late with invasion into the orbits and intracranium and compression of adjacent cranial nerves and their branches and hope this is now more acceptable to the reviewer.
- In the summary and introduction, it is stated that so far 100 cases were found, but authors perform study on 128 cases – it would be helpful to clarify this
Our reply: Thank you so much for your nice comment. We have modified the manuscript and change it to “less than 200 cases”. We sincerely hope that this will be acceptable to the reviewer.
- On the same note, in Table 2 and in whole chapter about histological findings authors describe 192 patients, but study includes only 128 – I suggest to include in this chapter/table only patients that are a part of the study this manuscript uses (128)
Our reply: Thank you very much for your valuable advice and nice suggestion. We have been well aware of this important and potentially controversial issue before we submitted the first manuscript. After careful consideration of the reviewer’s comment, we would like to keep the contents of Table 2 unchanged and include the histological findings of all 192 patients instead of 128 patients. The histological findings of all 192 patients (including 8 patients treated at our institution) were available from our systematic review. If we just include 128 patients and omit the histological findings of the remaining 64 patients, we will miss such important information in one-third of patients identified in our systematic review, which is not a negligible patient number. In that case, it will not reflect and represent an accurate distribution of the histological types and their subtypes in this systematic review and will miss some rarer histological subtypes. It will also probably raise challenges and concerns by other reviewers and future readers. We sincerely hope that our wish to illustrate the histological types and subtypes of 192 patients which are clearly identified in our systematic review is acceptable to the reviewer. Thank you very much for your kind consideration.
- In table 1 it would be helpful to define meaning of different categories listed in the table
Our reply: Thank you very much for your reply. We have provided the definition of T and N categories listed in Table 1 in a newly created Supplementary Table S1 in the Supplementary materials, so that this will not make Table 1 more difficult and complex to read. We hope this additional note is acceptable to the reviewer.
- Although authors included relatively high number of their own cases (8) they were collected in a period that end in March 2022 - cases diagnosed in last few (e. g. 6) months can hardly have useful follow up
Our reply: Thank you very much for your nice comments. As shown in the Result section of our manuscript, the median follow-up duration of the cohort of our 8 patients was 30.8 months (range 8.9–50.1), which was longer than that for the 120 patients in previous publication (median 17.0 months, ranging from 0.3 to 149.0 months). Only two patients had follow-up duration of less than 2 years (8.9 and 10.8 months, respectively) as they died of cerebral infarction and intradural spinal metastases respectively, while the follow up duration for the remaining 6 alive patients ranged from 21.4 months to 60.1 months. The description of inclusion of patients in our original manuscript may not be accurate enough. In this regard, we modified in our manuscript that our 8 patients treated between January 2014 and December 2021 were included with the dataset locked on 31st March 2022 for our subsequent survival analyses. We hope our elaboration and revision of our manuscript is now more acceptable to the reviewer.
- OS was defined by time between diagnosis and death by any causes – I strongly suggest to define it as period between diagnosis and death related to SMARCB1 (INI-1)-Deficient Sinonasal Carcinoma
Our reply: Thank you very much for your nice comment. We think the Reviewer is referring to cancer-specific survival (CSS) in the comment. We have reviewed all the 128 patients’ treatment details and survival outcomes again in our systematic review and found that only 1 patient died of causes unrelated to the SMARCB1-deficient sinonasal carcinoma. Furthermore, the cause of death of another 11 patients (shown in Supplementary Table S2 in our revised manuscript) was undetermined despite our further enquiry with emails sent to the corresponding authors of these publications. Since the cause of death of these 11 patients (9% of 128 patients in this systematic review) was not determined and only 1 patient died of other cause, it will be misleading to the reviewers and the readers if we omit these 11 patients in CSS analysis. The omission of the remaining 1 patient who died of other cause would not significantly affect the result in CSS analysis if it is performed. Therefore, we opine that overall survival would be the most appropriate survival end point in this systematic review, given the very limited information of survival outcomes of the included publications which mainly focus on the histological and genetic aspects with little description of the clinical course and survival outcomes. We sincerely hope that our explanation and elaboration is acceptable to the reviewer.
- It would also be beneficial to have information how was, in different studies, SMARCB1 deficiency established and if deletion was biallelic
Our reply: Thank you very much for your nice comment and valuable suggestions. We have added Section 3.3 in our revised manuscript to describe the deletional status of SMARCB1. The tumors of all 128 patients in this systematic review were confirmed to have loss of expression of SMARCB1 (INI-1) with immunohistochemistry. Additional genetic/genomic diagnostic tests were performed to confirm the status of SMARCB1 deletion in 24 patients: fluorescence in-situ hybridization (FISH) in 9 patients, next generation sequencing (NGS) in 2 patients (including one from our institution), NGS and FISH in 1 patient, NGS and chromogenic in-situ hybridization in 4 patients, and NGS and multiplex ligation-dependent probe amplification in 8 patients. Out of these 24 patients, monoallelic deletion was identified in 3 patients and biallelic deletion was found in 19 patients. Normal results of FISH were noted in the remaining 2 patients. We have updated this information in our revised manuscript, and hope that it will be acceptable and satisfactory to the reviewer.
- I don't see reason for the inclusion of phase II trial initiation in this manuscript – it doesn't add anything to the topic of the study since it is not jet performed
Our reply: Thank you very much indeed for your nice comment and suggestion. On the other hand, the inclusion of our proposed phase II trial on tazemetostat with induction chemotherapy and concurrent chemoradiation was well commended and praised by Reviewer 1. We have carefully considered this before the manuscript was submitted and think that it is still worthwhile to include our phase II trial in this manuscript. We are well aware that there have been more and more SCI-indexed journals having published the protocols of ongoing clinical trials which are still in progress. This will give a clearer direction and picture to the readers that there are ongoing studies with targeted therapies like tazemetostat which may improve the treatment and survival outcomes of patients who unfortunately suffer from this dreadful malignancy. We sincerely hope that our elaboration and the wish to keep the description of our ongoing trial in our manuscript is now more acceptable to the reviewer.
We are most grateful and appreciated once again to the reviewer's expert review, and hope that our elaboration and manuscript revision is now acceptable to the reviewer.